# Neuroprotective Effects of Erinacine A on an Experimental Model of Traumatic Optic Neuropathy

**DOI:** 10.3390/ijms24021504

**Published:** 2023-01-12

**Authors:** Chiao-Ling Hsu, Yao-Tseng Wen, Tzu-Chao Hsu, Chin-Chu Chen, Li-Ya Lee, Wan-Ping Chen, Rong-Kung Tsai

**Affiliations:** 1Institute of Medical Sciences, Tzu Chi University, Hualien 970, Taiwan; 2Institute of Eye Research, Hualien Tzu Chi Hospital, Buddhist Tzu Chi Medical Foundation, Hualien 970, Taiwan; 3Department of Medical Education, Medical Administration Office, Hualien Tzu Chi Hospital, Buddhist Tzu Chi Medical Foundation, Hualien 970, Taiwan; 4Biotech Research Institute, Grap King Bio Ltd., Taoyuan 325002, Taiwan; 5Doctoral Degree Program in Translational Medicine, Tzu Chi University and Academia Sinica, Hualien 970, Taiwan

**Keywords:** traumatic optic neuropathy, *Hericium erinaceus*, Erinacine A, antiapoptosis, anti-inflammation, oxidative stress, antioxidant, pRIP, Nrf2

## Abstract

Erinacine A (EA), a natural neuroprotectant, is isolated from a Chinese herbal medicine, *Hericium erinaceus*. The aim of this study was to investigate the neuroprotective effects of EA in a rat model of traumatic optic neuropathy. The optic nerves (ONs) of adult male Wistar rats were crushed using a standardized method and divided into three experimental groups: phosphate-buffered saline (PBS control)-treated group, standard EA dose-treated group (2.64 mg/kg in 0.5 mL of PBS), and double EA dose-treated group (5.28 mg/kg in 0.5 mL of PBS). After ON crush, each group was fed orally every day for 14 days before being euthanized. The visual function, retinal ganglion cell (RGC) density, and RGC apoptosis were determined using flash visual-evoked potentials (fVEP) analysis, retrograde Fluoro-Gold labelling, and TdT-dUTP nick end-labelling (TUNEL) assay, respectively. Macrophage infiltration of ON was detected by immunostaining (immunohistochemistry) for ED1. The protein levels of phosphor-receptor-interacting serine/threonine-protein kinase1 (pRIP1), caspase 8 (Cas8), cleaved caspase 3 (cCas3), tumour necrosis factor (TNF)-α, tumour necrosis factor receptor1 (TNFR1), interleukin (IL)-1β, inducible nitric oxide synthase (iNOS), nuclear factor erythroid 2-related factor 2 (Nrf2), haem oxygenase-1 (HO-1), and superoxide dismutase 1 (SOD1) were evaluated by Western blotting. When comparing the standard EA dose-treated group and the double EA dose-treated group with the PBS-treated group, fVEP analysis showed that the amplitudes of P1–N2 in the standard EA dose group and the double EA dose-treated group were 1.8 and 2.4-fold, respectively, higher than that in the PBS-treated group (*p* < 0.05). The density of RGC in the standard EA dose-treated group and the double EA dose-treated group were 2.3 and 3.7-fold, respectively, higher than that in the PBS-treated group (*p* < 0.05). The TUNEL assay showed that the standard EA dose-treated group and the double EA dose-treated group had significantly reduced numbers of apoptotic RGC by 10.0 and 15.6-fold, respectively, compared with the PBS-treated group (*p* < 0.05). The numbers of macrophages on ON were reduced by 1.8 and 2.2-fold in the standard EA dose-treated group and the double EA dose-treated group, respectively (*p* < 0.01). On the retinal samples, the levels of pRIP, Cas8, cCas3, TNF-α, TNFR1, IL-1β, and iNOS were decreased, whereas those of Nrf2, HO-1, and SOD1 were increased in both EA-treated groups compared to those in the PBS-treated group (*p* < 0.05). EA treatment has neuroprotective effects on an experimental model of traumatic optic neuropathy by suppressing apoptosis, neuroinflammation, and oxidative stress to protect the RGCs from death as well as preserving the visual function.

## 1. Introduction

Traumatic optic neuropathy (TON) is caused by accidents such as head trauma [1,2]. TON may cause permanent visual loss because of primary lesions and secondary degeneration. The secondary degeneration-associated factors include optic nerve (ON) edema, inflammatory cells infiltration, and barrier breakdown [2,3,4]. After the axonal injury, the myelin sheath of the axon lesions degenerates, which triggers the infiltration of ED-1-positive macrophages and induces inflammatory response [5,6]. In addition, the apoptotic cascade, such as caspases 3 (Cas3), is activated and, eventually, causes RGC death apoptosis and vision loss [7]. There are no effective treatments established for TON, but optic canal decompression and use of corticosteroids may have limited therapeutic application compared to observations alone [8,9,10]. ON is part of CNS, for which the search for the effective treatment for TON to preserve visual function is emerging.

The molecular pathogenesis of RGC death in TON is complicated. RIP1-dependent apoptosis (RDA) was reported to lead RGC death in an experimental model of TON [11]. Receptor-interacting serine/threonine-protein kinase 1 (RIP1) manipulates certain biological functions, such as apoptosis, necroptosis, and inflammation [12]. Activation of RIP1 affects the RIP1-dependent apoptosis (RDA), and the inflammatory responses have been reported in various human neurodegenerative diseases [13,14]. Previous reports demonstrated that RIP1 activation promotes the levels of inflammatory genes and releases pro-inflammatory cytokines such as tumour necrosis factor-α (TNF-α) [15,16]. Besides, TNF-α interacts with tumour necrosis factor receptor1 (TNFR1) and triggers the activation of downstream Caspase 8 (Cas8), promoting the apoptosis cascade [12]. After ON injury, RGCs go through caspase-mediated apoptosis; especially, Cas8 is a key factor involved in apoptosis and necroptosis [17].

A cellular balance exists between oxidative stress and antioxidative defence mechanisms to maintain redox homeostasis [18,19]. After an axonal injury, the intracellular production of superoxide induced oxidative stress, leading to RGC death [20,21]. Oxidative stress was one of the mechanisms in TON that caused neurodegeneration [4,21]. In a murine-controlled cortical impact over the visual cortex model, oxidative stress involved RGC apoptosis and ON degeneration through the inhibition of nuclear factor erythroid 2-related factor 2 (Nrf2) [21,22]. The Nrf2 protein is a transcription factor for promoting antioxidant/detoxification gene transcription. With excessive ROS generation, Nrf2 translocates into the nucleus, and nuclear Nrf2 binds to the antioxidant response element for transcription antioxidant genes such as haem oxygenase-1 (HO-1) [23,24]. Increased oxidative stress and RGC death after ON crush, in the Nrf2 knockout (KO) mice model, has been reported [25,26,27]. Moreover, Nrf2 overexpression prevented RGC apoptosis and promoted antioxidant gene expression in the ON crush model [28]. The neuroprotective effect of serum response factor reduced RGC apoptosis with high-glucose insult by regulating Nrf2 [29]. Monomethyl fumarate, improving Nrf2 target gene expression, could inhibit inflammatory response, increased neuronal cell survival, and recovered electrophysiological function after ischemia/reperfusion (I/R) injury in Nrf2 KO mice [30]. Another antioxidant enzyme is Cu/Zn-superoxide dismutase (Cu/Zn-SOD, SOD1), which converts superoxide to O_2_ and H_2_O_2_ [19,31]. Oxidative stress is implicated in many neurodegenerative diseases, such as Nrf2 destabilization in Parkinson’s Disease (PD) and *SOD1* mutation in Amyotrophic Lateral Sclerosis (ALS) [32]. Similarly, superoxide is an upstream signal for retinal ganglion cell apoptosis after optic nerve injury [33,34].

*Hericium erinaceus* (Lion’s trade mane mushroom, yamabushitake, and Houtou) is a kind of mushroom that is used in traditional Chinese medicine for the treatment of gastritis and other digestive tract-related diseases in East Asian countries [35,36]. The fruiting bodies and mycelium of *H. erinaceus* were isolated, and its function is associated with anti-inflammatory, antiproliferative, antimicrobial, and neuritogenic activities [35,37,38]. *H. erinaceus* increases nerve growth factor (NGF) protein expression through the JNK pathway in 1321N1 human astrocytoma cells [39]. The neuroprotective capabilities of *H. erinaceus* prevent neurological degeneration, such as AD and PD [40]. In the APPswe/PS1dE9 transgenic mice model of AD, *H. erinaceus* mycelia (HE-My) and ethanol extracts of HE-My not only increase NGF/proNGF ratio but also improve Aβ plaques and microglia-mediated clearance [41]. The fruiting bodies of *H. erinaceus* have been reported to have antioxidant effects and antiosteoporotic activities [42]. In the renal I/R mouse model, I/R increased renal oxidative stress and inhibited antioxidant enzyme (glutathione) activity. Pretreatment with *H. erinaceus* polysaccharide in mice downregulated lipid peroxidation levels and upregulated antioxidant enzyme activities [43]. In HT22, the mouse hippocampal neurones and lipopolysaccharide (LPS)-induced BV-2 microglial cells experiment, *H. erinaceus* reduces ROS production and increases the antioxidant enzyme catalase and glutathione [44].

Erinacine A (EA), a cyathane diterpenoid, was isolated from the MeOH extract of the cultured mycelia of *H. erinaceus*. EA, Erinacine B, and C (EC) are representative small molecules that promote NGF secretion into the cultured medium in mouse astroglial cells [45]. Moreover, EA stimulates catecholamine production, which increases NGF synthesis in the central nervous system of rats [46,47]. In an in vitro study, the neuroprotective and neuritogenic effects of EA promoted neurite outgrowth by activating tyrosine kinase A and Erk1/2 in pheochromocytoma (PC12) cells [48]. In the 1-methyl-4-phenyl-1,2,3,6-tetrahydropyridine (MPTP) disease animal model, EA affects the expression of Fas and Bax through IRE1α/TRAF2 complex formation and the phosphorylation of JNK1/2, p38, and NF-κB pathways, inhibiting endoplasmic reticulum stress and improving neuronal survival [49]. In addition, post-treatment with EA reduces not only neurotoxicity, by interfering with the IRE1α/TRAF2 complex formation, but also activation of PAK1, AKT, LIMK2, ERK, and Cofilin, which improve neuronal survival in the MPTP model of PD [50]. EA reduces neuron cell apoptosis by inhibiting the Bax/Bcl-2 ratio and Cas3 (ROS-caspase dependent pathway) in vitro [51], as well as causing a reduction in ROS and inflammatory cytokines in vivo [52]. 

EA, the main representative of this compound group, is one of the stimulators of NGF synthesis, but its anti-inflammatory and anti-apoptotic functions in TON are not yet confirmed. Thus, this study aimed to investigate the neuroprotective effects of EA applications after the ON crush injury model in rats and related molecular mechanisms.

## 2. Results

### 2.1. Treatment with EA Preserves Visual Function

To assess visual function, we evaluated the sham group, Crushed+phosphate-buffered saline-treated (C+PBS-treated), Crushed+standard EA dose-treated, and Crushed+double EA dose-treated groups by ash visual-evoked potentials (fVEP). In comparison with the sham group, the P1 latency did not show significant variance in the C+standard EA dose-treated group and the C+double EA dose-treated group (Figure 1A). The amplitudes of the P1–N2 waves in the sham, C+PBS-treated, C+standard EA dose-treated, and C+double EA dose-treated groups were 59.2 ± 19.3, 16.8 ± 5.1, 30.6 ± 8.9, and 39.5 ± 25.9 µV, respectively (* *p* < 0.05, ** *p* < 0.01; Figure 1B). The P1–N2 amplitudes were 1.8 and 2.4-fold higher in the C+standard EA dose-treated group and the C+double EA dose-treated group, respectively, than in the C+PBS-treated group.

### 2.2. Treatment with EA Enhances RGC Survival Rate and Reduces RGC Apoptosis

To evaluate the effect of EA on RGC survival, we calculated the density of RGCs in the central retinas (Figure 2A). The sham, C+PBS-treated, C+standard EA dose-treated, and C+double EA dose-treated groups had densities of 2352 ± 90.4, 345.3 ± 180.6, 780.1 ± 403.0, and 1286 ± 284.5/mm^2^, respectively (* *p* < 0.05, *** *p* < 0.001; Figure 2B). The RGC densities in the C+standard EA dose-treated group and the C+double EA dose-treated group were 2.3 and 3.7-fold higher than that in the C+PBS-treated group, respectively.

The numbers of TdT-dUTP nick end-labelling (TUNEL)-positive cells in the retinal ganglion cell layer (GCL) of the sham, C+PBS-treated, C+standard EA dose-treated, and C+double EA dose-treated groups were 0.3 ± 0.3, 11.1 ± 5.7, 1.1 ± 0.8, and 0.7 ± 0.3/high-power field (HPF), respectively (* *p* < 0.05, ** *p* < 0.01; Figure 3B). The C+standard EA dose-treated and C+double EA dose-treated groups had significantly reduced numbers of apoptotic RGC by 10.0 and 15.6-fold, respectively, compared with the C+PBS-treated group.

### 2.3. Treatment with EA Avoids Macrophage Infiltration in the on Tissue

After ON crush, the lesion site of ON showed macrophage infiltration. To evaluate the anti-inflammatory efficacy of the EA, we used the ED1 (CD68) antibody for the immunostaining of activated macrophages (Figure 4A). The numbers of ED1-positive cells/HPF in the sham, C+PBS-treated, C+standard EA dose-treated, and C+double EA dose-treated groups were 1.2 ± 1.2, 21.8 ± 4.4, 12.1 ± 5.0, and 10.0 ± 3.1, respectively (** *p* < 0.01, **** *p* < 0.0001; Figure 4B). The C+standard EA dose-treated group and the C+double EA dose-treated group reduced ED1-positive cells 1.8 and 2.2-fold, respectively, compared with the C+PBS-treated group in the ON crush model.

### 2.4. Treatment with EA Inhibits Apoptosis, Decreases Inflammation, and Enhances Antioxidative Stress Ability

To investigate the antiapoptosis and anti-inflammatory mechanisms of EA, we analysed the expression levels of target proteins in retinal samples through Western blotting. The C+double EA dose-treated group reduced the levels of pRIP, Cas8, and cCas3 by 2.2-fold (* *p* < 0.05), 2.1-fold (** *p* < 0.01), and 3.9-fold (** *p* < 0.01), respectively, compared with the C+PBS-treated group. The C+double EA dose-treated group decreased the levels of pRIP1 by 2.0-fold (* *p* < 0.05), respectively, compared with the C+standard EA dose-treated group (Figure 5A,B). EA treatment effectively inhibited RDA in retinal tissue after ON crush.

The C+standard EA dose-treated group inhibited the levels of inflammatory cytokines TNF-α, IL-1β, and iNOS by 2.3-fold (** *p* < 0.01), 2.0-fold (* *p* < 0.05), and 5.2-fold (* *p* < 0.05), respectively, compared with the C+PBS-treated group. The C+double EA dose-treated group downregulated the levels of inflammatory cytokines TNF-α, TNFR1, IL-1β, and iNOS by 2.0-fold (* *p* < 0.05), 1.7-fold (* *p* < 0.05), 1.9-fold (* *p* < 0.05), and 5.2-fold (* *p* < 0.05), respectively, compared with the C+PBS-treated group (Figure 6A,B). EA treatment significantly decreased the inflammatory response in ON tissue after ON crush.

Oxidative stress can activate cell apoptosis and play a crucial role in the TON pathogenesis [4,19]. We investigated the expression level of antioxidative stress proteins by Western blotting. The C+standard EA dose-treated group increased the levels of Nrf2 and SOD1 by 2.9-fold (* *p* < 0.05) and 2.1-fold (** *p* < 0.01), respectively, compared with the C+PBS-treated group. The C+double EA dose-treated group upregulated the levels of HO-1, and SOD1 by 1.9-fold (** *p* < 0.01), and 2.0-fold (* *p* < 0.05), respectively, compared with the C+PBS-treated group (Figure 7A,B). EA application promoted antioxidative stress ability in retinal tissues after ON crush.

## 3. Discussion

After the ON crash procedure, we gave two different dosages of EA for 14 days: 2.64 mg/kg in 0.5 mL of PBS and 5.28 mg/kg in 0.5 mL of PBS. In the ON crush model, fVEP data shows that EA treatment preserves visual function, whereas RGC counting and TUNEL assay data show that EA treatment decreases RGC apoptosis. Moreover, ED-1 immunostaining shows that EA treatment inhibits inflammatory response on ON. Finally, Western blotting shows that EA treatment reduces the expression of proinflammatory cytokines, suppresses the phosphorylation *of RIP1*, and strengthens the Nrf2/HO-1 antioxidant pathway.

EA deactivates oxidative stress-dependent CHOP expression for decreasing oxidative stress of MPP^+^-induced neuronal damage both in vitro and in vivo [50]. EA-enriched HE-My promotes longevity by decreasing levels of thiobarbituric acid reactive substances and inducing SOD in mice [53]. However, no study has demonstrated that EA enhances antioxidant capacity by activating the Nrf2/HO-1 pathway in TON. In addition, EC decreases Keap1 protein expression and increases nuclear Nrf2 protein expression, which activates the Nrf2/Ho-1 pathway in LPS-induced human BV-2 microglia cells [54]. Pretreatment with the *H. erinaceus* extract would activate the expressions of nuclear Nrf2, HO-1, γ-glutamylcysteine synthetase, and glutathione (GSH), thereby inhibiting the ROS induced by TNF-α in human endothelial (EA.hy926) cells [55]. The *H. erinaceus* extract contains EA, which may be involved in the activation of Nrf2 and the enhancement of the antioxidant capacity. Our previous studies also revealed the neuroprotective effects of the soluble *p*-selectin, through the activation of the Nrf2/HO-1/Nqo-1 pathway, in a rat model of anterior ischemic optic neuropathy [56], and intravitreal injection of long-acting, pegylated granulocyte colony-stimulating factor also upregulated the AKT/Nrf2/HO-1 expression to provide neuroprotective effects in a TON model [57]. In summary, to the best of our knowledge, this is the first study demonstrating that the neuroprotective effects of oral posttreatment of EA enhance the Nrf2/HO-1/SOD1 expression and activate the effect of antioxidative stress after ON crush.

EA not only improves antioxidative stress resistance but also suppresses inflammatory responses. With intraperitoneal administration of EA 24 h pre-stroke, EA reduced the serum protein expressions of proinflammatory cytokines iNOS, IL-1, IL-6, and TNF-α in the cerebral global ischaemic stroke animal model [52]. After feeding high-fat and high-sucrose diets mixed with Hericium erinaceus mycelium (HEM) or EA for 18 weeks, lowering hippocampal messenger RNA (mRNA) expressions of TNF-α and IL-1β, HEM and EA limit the progression of obesity-induced neurodegeneration [58]. EA prevents iNOS and NO production in LPS induced neurodegeneration both in vitro and in vivo [59]. Consistently, our observations also showed that post-treatment with EA by oral gavage decreased the TNF-α/TNFR1, IL-1β, and iNOS expressions in ON samples post-crush injury.

EA increases cell survival by inhibiting Bax/Bcl-2, Cas3, and NF-κB protein expressions in several in vitro models [49,51]. EA improves cell survival by increasing antioxidant capacity and decreasing inflammatory response through various mechanisms [48,49,50,51,52,53,59]. In this study, we demonstrated that EA decreases RGC death by inhibiting the RDA pathway in the TON model.

In a closed-globe blunt ocular injury model, the rats received intravitreal injection of Nec-1s (a RIP1 inhibitor) immediately and 7 days post-injury, and observations confirmed that Nec-1s protected RGCs from cell death at the centre of the injury site [11]. In a retinal I/R model, pre-treatment with an intravitreal injection of Nec-1 protected the inner nuclear layer and GCL from apoptosis and reduced the TNF-α-induced RIP1/RIP3 necroptosis [60]. Our observations showed a significant reduction in TNF-α, pRIP1, and Cas8 expressions after EA treatment post-ON crush. As mentioned above, RIP involves inflammatory response, apoptosis, and oxidative stress. EA is a natural product that probably contains RIP1 inhibitory components and, thus, provides ON protection.

EA is a natural product of cyathane diterpenes, and its mechanisms vary depending on the disease model. In short, cyathane diterpenes have direct effects on both the promotion and inhibition of neurogenesis [61]. EA’s multiple mechanisms not only stimulate NGF and brain-derived neurotrophic factor pathways but also disrupt mitochondrial function, resulting in excessive ROS production [62,63,64]. In colorectal cancer cells, EA induces the expressions of TNFR, Fas ligand, and FasR to activate intrinsic and extrinsic apoptosis pathways [65]. In gastric cancer cells, EA activates apoptosis through the FAK/AKT/PAK1 pathway [66].

Our study demonstrated that EA treatment enhances the Nrf2/HO-1/SOD1, represses TNF-α/TNFR1, IL-1β, and iNOS, and it reduces pRIP1/Cas8/cCas3 expressions to achieve the neuroprotective effects after TON insult.

## 4. Materials and Methods

### 4.1. Animals

Male Wistar rats (weighing 150–180 g, 6–8 weeks old) were purchased from the breeding colony of BioLASCO Co., Taiwan. The rats were provided free access to food (PMI5001, LabDiet) and water (reverse osmosis) and lived in cages with a room temperature of 23 ± 1 °C, humidity of 55 ± 5%, and a 12-h light/dark cycle. Animal care and experimental procedures followed the ARVO Statement for the Use of Animals in Ophthalmic and Vision Research. The Institutional Animal Care and Use Committee at Tzu Chi Medical Center approved all animal experiments. All manipulations were performed with animals under general anaesthesia, a cocktail of ketamine (100 mg/kg; Merial, France) and xylazine (10 mg/kg; Health-Tech Pharmaceutical Co., Taipei, Taiwan), by intramuscular injection. Local anaesthesia, Alcaine (Alocon Co., Puurs, Belgium), was applied, and pupil dilation, Mydrin-*p* (Santen Oy., Tempere, Finland), were performed in all experiments.

### 4.2. Study Design

In the present study, we investigated the neuroprotective effects of EA in a rat ON crush model (Figure 8). We used a total of 120 rats; of these rats, 30 received sham operation, and 90 were subjected to ON crush in the right eye and were randomized into three groups (*n* = 30, each group). The Sham group was treated with o.5 mL of PBS. Among the ON-crushed groups, one was treated with 0.5 mL of PBS, and the other two ON-crushed groups were treated with different dosages of EA. The safety dosage of the EA has been validated in the Sprague–Dawley rats, fed for 28 days at a maximum dose of 3 g/kg, followed by analysis systemic toxicity assays, including urinalysis, haematology, and serum biochemistry parameters. There were no pathological and histopathological findings [67]. In our previous data, a rat with a weight of about 200 g received 40 mg HE extract, and each mg of HE extract contained 66 ug of EA, so there was 2.64 mg of EA in the HE extract. Therefore, this experiment is based on this basis to perform EA on TON. In this study, two dosages by oral gavage were given with a concentration of 2.64 mg/kg in 0.5 mL of PBS (C+standard EA dose-treated) and 5.28 mg/kg in 0.5 mL of PBS (C+double EA dose-treated), respectively. To investigate the proteomic changes at early stages of post-injury and treatment, we collected protein samples (including retinas and ONs) for Western blot analysis at day 3 post-operation (*n* = 6 each group). At day 14 post-operation, visual function was assessed by fVEP analysis (*n* = 6 each group), RGC survivors were counted by retrograde Fluoro-Gold labelling (*n* = 6 each group), and RGC apoptosis in the GCL was detected through in situ TUNEL assay (*n* = 6 each group). Inflammatory infiltration of macrophages in the ON section was determined by using immunohistochemistry (IHC) (*n* = 6 each group).

### 4.3. ON Crush Model in Rats and EA Application

The procedure of ON crush was described in detail in our previous reports [68,69,70]. After general anaesthesia and topical Alcaine eye drop application, under an operating microscope, the ON of the right eye was exposed and isolated, which carefully avoided damaging the small vessels around the ON. At a distance of 2 mm posterior to the eyeball, a vascular clip (60-g micro-vascular clip; World Precision Instruments, Sarasota, FL, USA) was applied for 30 s, leading to ON injury on each rat’s right eye. After the operation, Tobradex eye ointment (Alcon, Puurs, Belgium) was administered. Subsequently, the rats were kept on electronic heating pads at 37 °C for recovery. In the sham control group of rats, the ON of the right eye was exposed without doing the crush experiments to the ON and serving as a normal control.

After ON crush, we performed oral gavage PBS or EA twice daily, from day 0 to day 14, at 7 am and 7 pm on the ON-crushed group, which followed the procedure on the Institutional Animal Care and Use Committee at Tzu Chi Medical Center’s approval. The sham group received PBS gavage only and served as a normal control. The extracted EA has a purity of 99.0396% and was provided by Biotech Research Institute (Grap King Bio Ltd., Taoyuan, Taiwan). The feeding procedures were as follows: 1. Choose 15–18 G, 6–8 cm metal gavage needle and measure weight of the rats and the volume (0.5 mL/rat) to be administered. 2. Measure the distance the tube is into the oesophagus from the mouth to the last rib. 3. Pre-fill the syringe and gavage tube with the PBS/EA to be performed, and exclude any air out of the gavage tube. 4. Restraint of the head, neck, and upper limb of the rats ensures the rats can breathe freely. 5. Following the curve of the oropharynx, insert the gavage tube into the left side of the rat’s mouth in the gap. First of all, the slight resistance you will feel comes from the larynx. At this time, the rat will make a gagging reaction as you go through the larynx to the oesophagus. It will feel easy to move forward without pressure. 6. The rats are observed for respiration by visual checking and are injected with a small amount of PBS/EA first to avoid chocking episodes. If the respiration remains normal, the PBS/EA is fully injected. 7. The gavage tube is slowly removed, and the rat is returned to the cage for observation, for approximately 10 min, to confirm no complications.

### 4.4. fVEPs

We used a visual electrodiagnostic system (Espion, Diagnosys LLC, Gaithersburg, MA, USA) to measure the fVEPs 2 weeks after the operation, and the detailed processes have been described in previously published studies [6,68,69,71]. All rats had suffered the ON of the right eye exposed with/without crush. Due to the closed distance between the two eyes of the rat, there was a concern regarding the interference of the other eye during the FVEP measurement. For the fVEP examinations, we performed the same ON crush surgery in both eyes to avoid VEP responses of albino Wistar rats being contaminated from the contralateral side. According to the stereotaxic coordinate method suggested by Ohlsson et al., after incising the skin of the skull and finding the bregma, the frontal cortex area (negative electrode) was 8 mm away from the bregma, and the primary visual cortex area (positive electrode) was 3 mm on both sides of the bregma, respectively [72].We collected electrodes of the frontal cortex region as reference data and electrodes of the primary visual cortex region as active data. We defined the first positive wavelet as the P1 wave and the first negative wavelet as the N1 wave. The amplitudes of the P1–N2 waves were compared among the groups to estimate visual function (*n* = 6 each group).

### 4.5. Retrograde Labeling of RGCs with Fluoro-Gold

We described the detailed methods and protocols in previous reports [68,69,70,73]. On day 7 after the operation, retrograde labelling of the RGCs was performed; 2 µL of 5% Fluoro-Gold (Fluorochrome LLC, Denver, CO, USA) was injected into the superior colliculus, at a depth of 4.5 mm from the surface of the skull, by a Hamilton syringe. The dye was injected at a point 5.5 mm caudal to the bregma and 1.5 mm lateral to the midline on both sides. After injection, we filled the holes with bone wax and sutured the incision. One week after the labelling, the eyeballs were harvested for retina preparation after euthanasia. After the eyeballs were soaked in 10% formaldehyde for 1 h, the retinas were dissected and flattened by our radial cuts and mounted, with the vitreous side up, on a microscope slide. We calculated the central RGC densities at distances of 1 mm from the ON head through a 400 epifluorescence microscope (Axioskop; Carl Zeiss Meditech Inc., Thornwood, NY, USA) and a digital imaging system. The number of RGCs in five randomly selected areas (62,500 µm^2^) was counted and calculated by ImageMaster 2D Platinum software (*n* = 6 each group).

### 4.6. TUNEL Assay for Apoptotic Cell Measurements

The principle of TUNEL assay (DeadEnd^TM^ Fluorometric TUNEL System; Promega Corporation, Madison, WI, USA) is labelling the TdT-dUTP terminal nick positive cells. We used this to detect apoptotic cells in the GCL, and it was performed according to the manufacturer’s procedure. All frozen sections of the retina samples were prepared, at distances of 1–2 mm from the ON head, to ensure the use of equivalent fields for comparisons. Each retina sample was counted by 10× HPF (400× magnification) (*n* = 6 each group).

### 4.7. IHC of ED1

ED1 (CD68) is a marker of activated microglia/macrophages [68,69,74]. Briefly, the frozen ON sections were fixed on a 37 °C dry bath for 10 min, and they were blocked with 3% BSA for 1 h. The ED1 primary antibody (1:100; Abcam, San Francisco, CA, USA) was incubated with the ON section overnight at 4 °C. On the next day, the secondary antibody conjugated with fluorescein isothiocyanate (1:500; Jackson ImmunoResearch Laboratories, West Grove, PA, USA) was applied at room temperature of 23℃ for 1 h, and counterstaining was performed using DAPI (1:1000; Sigma, St. Louis, MO, USA), which expressed the nucleus of ON cells. Then, we counted the ED1-positive cells of the ON lesion sites in 6 HPF (400× magnification) (*n* = 6 each group).

### 4.8. Western Blotting 

On day 3 post-operation, the retina and ON samples were harvested, homogenized, and quantified. Then, 20 µg of proteins was separated by a 4–12% NuPAGE Bis–Tris gel (Thermo Scientific, Rockford, IL, USA) and transferred onto polyvinylidene difluoride (PVDF) membranes by iBlot 2 Gel Transfer Device (Thermo Scientific). Then, PVDF membranes were blocked with 5% BSA for 1 h and were incubated with pRIP1 (Ser166) (1:1000, no. 65746; Cell Signaling Technology Inc., Danvers, MA, USA), Cas8 (1:1000, no. 4790; Cell Signaling Technology Inc., Danvers, MA, USA), cleaved Caspase 3 (1:1000, no. 9662; Cell Signaling Technology Inc., Danvers, MA, USA), TNF-α (1:1000, ab10863; Abcam), IL-1β (1:1000, ab9722; Abcam), TNFR1 (1:1000, ab19139; Abcam), iNOS (1:1000, ab3523; Abcam), Nrf2 (1:1000, ab137550; Abcam), HO-1 (1:1000, ab68477; Abcam), SOD1 (1:1000, ab13498; Abcam), and GAPDH (1:10,000, no. 2118S; Cell Signaling Technology Inc., Danvers, MA, USA) primary antibodies, overnight, at 4 °C. After washing, PVDF membranes were incubated with secondary antibodies, conjugated to horseradish peroxidase, against the appropriate host species (1:10,000; Bio-Rad Laboratories, Inc., Hercules, CA, USA) for 1 h at a room temperature of 25 °C. Subsequently, we used an enhanced chemiluminescent substrate (Perkin-Elmer Life Science, Boston, MA, USA) and Thermo Fisher’s iBright Analysis software (Thermo Fisher Scientific) for detecting protein bands. The results were normalized with GAPDH, and the statistical analysis was performed by the Kruskal–Wallis test (*n* = 6 each group).

### 4.9. Statistical Analysis

In the present study, each measurement was performed in a blinded fashion. All statistical analyses were evaluated using a GraphPad Prism. The Kruskal–Wallis test was used to compare the groups. Data are presented as mean ± SD. A *p*-value < 0.05 was considered statistically significant.

## 5. Conclusions

In summary, treatment with EA provides the neuroprotective effects on RGCs morphometry and preserves the visual function in an animal model of TON, as evidenced by RGC survival and fVEP. The main protective effects of EA on an ON-crush model might occur through the actions of anti-apoptosis, anti-inflammatory response, and antioxidative stress (Figure 9).

## Figures and Tables

**Figure 1 ijms-24-01504-f001:**
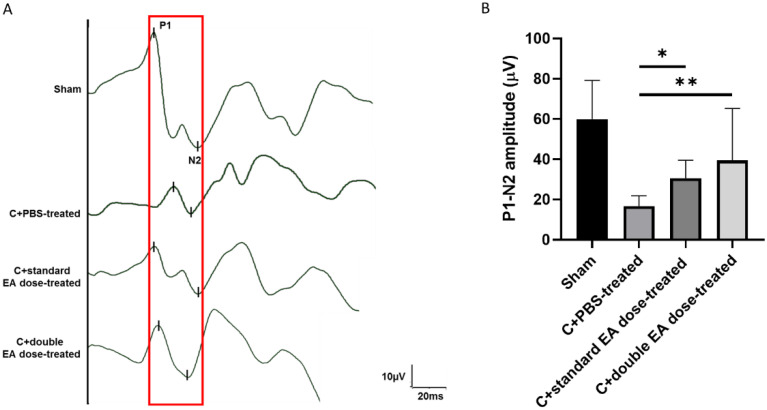
Evaluation of the visual function using fVEPs in the ON crush model. (**A**) Representative fVEP tracings at 2 weeks after ON crush in the sham, C+PBS-treated, C+standard EA dose-treated, and C+double EA dose-treated groups. Red box is the highlight of P1-N2 amplitude in fVEPs. (**B**) Bar charts demonstrate the P1–N2 amplitude. The amplitude values are expressed as mean ± standard deviation (SD) in each group (*n* = 6; * *p* < 0.05, ** *p* < 0.01).

**Figure 2 ijms-24-01504-f002:**
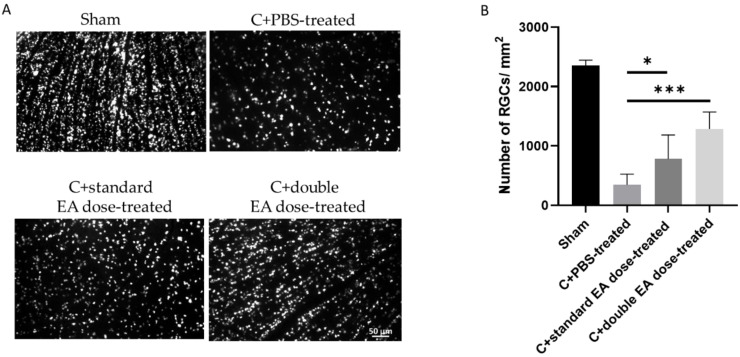
Survival of RGCs in the C+PBS-treated, C+standard EA dose-treated, and. C+double EA dose-treated groups at 14 days post-ON crush. (**A**) Representative of flat-mounted central retinas and the morphometry of RGCs in each group, by Fluoro-Gold retrograde labelling, at 2 weeks after ON crush. (**B**) RGC density of the central retinas in each group. Data are expressed as mean ± SD for each group (*n* = 6; * *p* < 0.05, *** *p* < 0.001).

**Figure 3 ijms-24-01504-f003:**
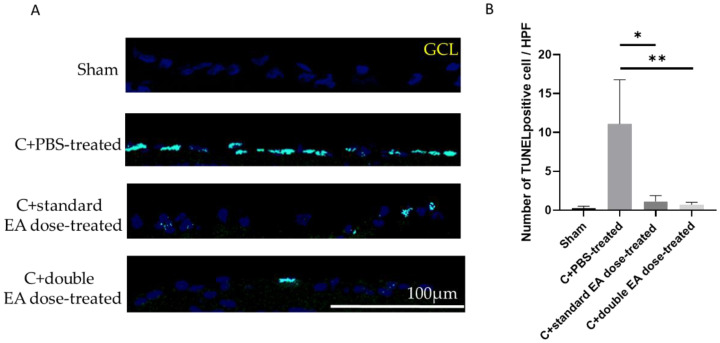
Analysis of RGC apoptosis in the GCL, through the TUNEL assay, at 2 weeks after ON crush. (**A**) Representative images of double-stained apoptotic RGCs in each group. The apoptotic cells (TUNEL-positive cells) in green were stained with TUNEL, and the nuclei of the RGCs in blue were labelled with DAPI staining. (**B**) Quantification of TUNEL-positive cells per HPF. Data are expressed as mean ± SD for each group (*n* = 6; * *p* < 0.05, ** *p* < 0.01).

**Figure 4 ijms-24-01504-f004:**
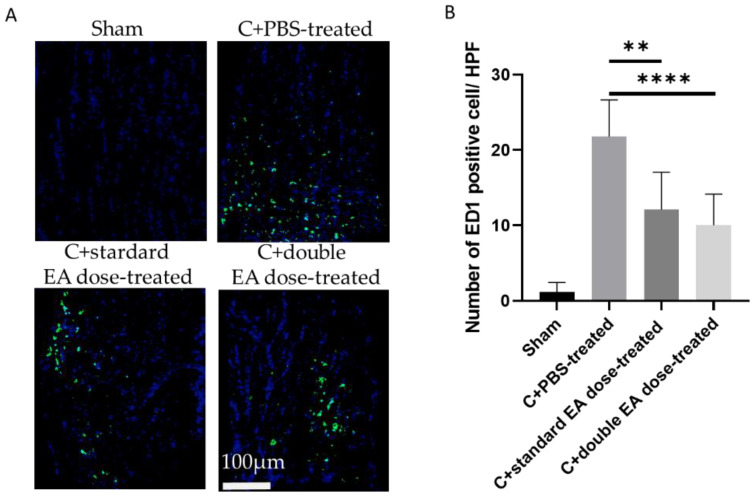
ED1 immunostaining of the ON section for evaluating inflammatory infiltration of macrophages. (**A**) Representative images of double-stained ON sections in each group. The macrophage-positive cells in green were stained with ED1, and the nuclei of the ON in blue were labelled with DAPI staining. (**B**) Quantification of ED1-positive cells per high-power field. Data are expressed as mean ± SD in each group (*n* = 6; ** *p* < 0.01, **** *p* < 0.0001).

**Figure 5 ijms-24-01504-f005:**
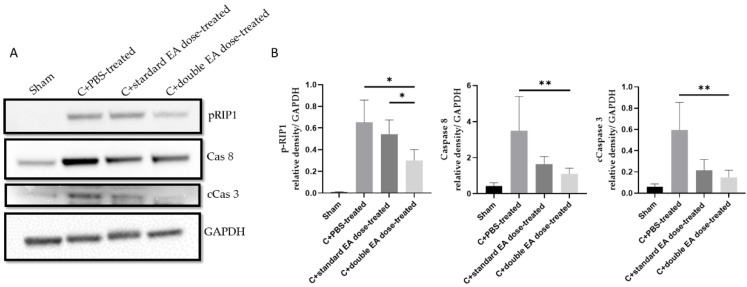
Levels of apoptosis proteins pRIP1, Cas8, and cCas3 in retinal tissues. (**A**) Analysis of the protein expression levels of pRIP, Cas8, and cCas3 by Western blotting. (**B**) Quantification of the protein bands. Data are expressed as mean ± SD in each group (*n* = 6; * *p* < 0.05, ** *p* < 0.01).

**Figure 6 ijms-24-01504-f006:**
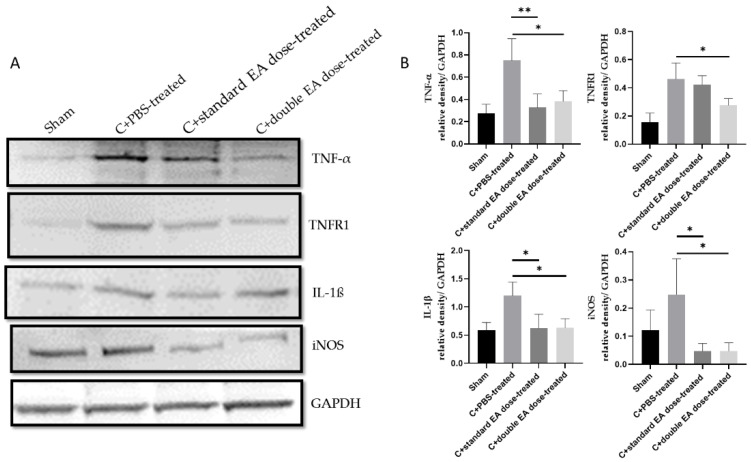
Levels of inflammation cytokines TNF-α, TNFR1, IL-1β, and iNOS in ON tissue. (**A**) Analysis of the protein expression levels of TNF-α, TNFR1, IL-1β, and iNOS by Western blotting. (**B**) Quantification of the protein bands. Data are expressed as mean ± SD in each group (*n* = 6; * *p* < 0.05, and ** *p* < 0.01).

**Figure 7 ijms-24-01504-f007:**
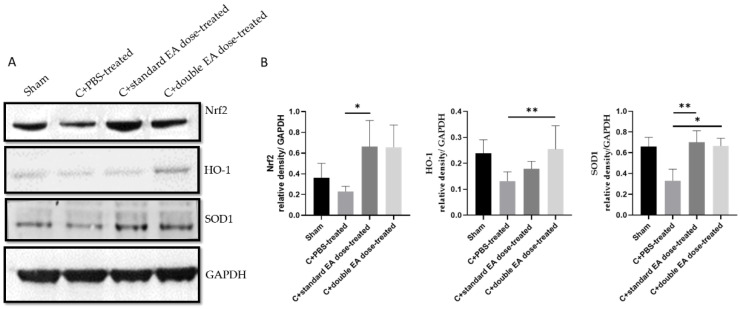
Levels of antioxidative stress proteins Nrf2, SOD1, and HO-1 in retinal tissues. (**A**) Analysis of the protein expression levels of Nrf2, HO-1, and SOD1 by Western blotting. (**B**) Quantification of the protein bands. Data are expressed as mean ± SD in each group (*n* = 6; * *p* < 0.05, and ** *p* < 0.01).

**Figure 8 ijms-24-01504-f008:**
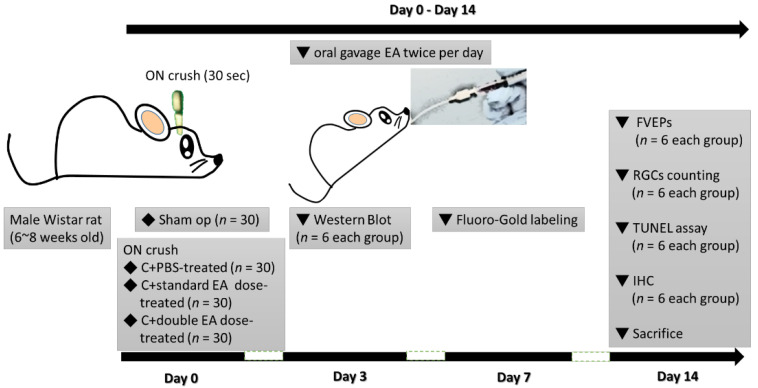
Description of the study design to investigate the role of EA on ON crush in rats.

**Figure 9 ijms-24-01504-f009:**
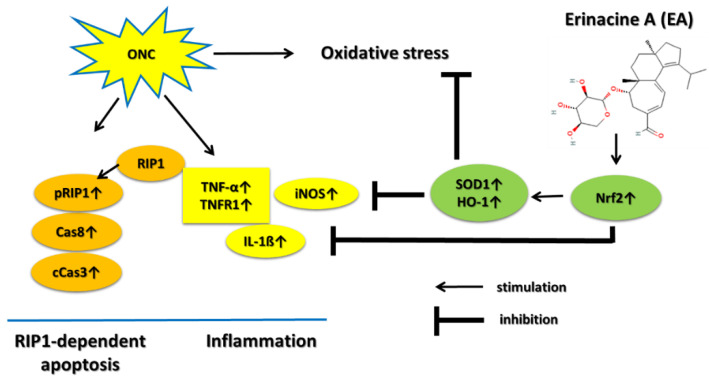
Mechanisms of the neuroprotective effects of EA in a rat ON-crush model (ONC). EA treatment activates Nrf2/HO-1/SOD1 antioxidative stress, inhibits TNF-α/TNFR1/IL-1β/iNOS inflammatory response, and reduces pRIP1/Cas8/cCas3 RIP1-dependent apoptosis.

## Data Availability

The data presented in this study are available in the article.

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
