# Peer review of "Neuroprotective Effects of Erinacine A on an Experimental Model of Traumatic Optic Neuropathy"

_ijms, 2023, doi:10.3390/ijms24021504_

Round 1
Reviewer 1 Report (Previous Reviewer 2)
All my previous comments have been addressed.
Reviewer 2 Report (Previous Reviewer 3)
I don't have further comments. The authors have addressed the questions from academic editors.
This manuscript is a resubmission of an earlier submission. The following is a list of the peer review reports and author responses from that submission.
Round 1
Reviewer 1 Report
General comments:
This manuscript reports the effect of Erinacine B on traumatic optic neuropathy. However, there are major concerns that should be addressed seeing as follows:
Major Comments:
1. In method section, the different sampling time points were indicated in the different groups. At day 3 post-operation, the samples were collected for protein detection; at day 14 post-operation, the samples were collected for other experiments. Why were different time points chosen? Is there any research supporting this method?
2. In Figure 4, the staining of ED1 appeared to be nonspecific in tissue sections. Please provide the picture of negative control.
3. In Figure 7, the band of Nrf2 showed more protein levels than the loading control of housekeeping gene. Was there the same quantity of proteins loaded in western blotting? The author should check the data.
4. Authors have provided very fewer description in the conclusion. It should be elaborated based on outcomes.
Author Response
1. In method section, the different sampling time points were indicated in the different groups. At day 3 post-operation, the samples were collected for protein detection; at day 14 post-operation, the samples were collected for other experiments. Why were different time points chosen? Is there any research supporting this method?
Response1:
Thanks for your valuable suggestion. TON induces RGC death gradually. RGC survival was 47% at 7dpi (day post injury) and 27% at 14dpi later [1]. Following optic nerve injury and subsequent Wallerian degeneration, it was found decrease of degenerating axons between 2 to 7 dpi. These degenerative events were paralleled by an intense glial cell reaction. Macrophages appear to be responsible for clearing most of the myelin debris from CNS lesions, peaking at 7 to 14dpi [2, 3]. In short, the RGC damage caused by TON, the rapid death of RGC peaked at 2-7 dpi, and after 14 dpi, whether apoptosis or macrophage participation tended to a smooth and relatively unchanged period. Therefore, we evaluated the western blot of ON at 3 days post injury, and FVEP, TUNEL assay and Fluoro-Gold to evaluate RGC apoptosis, and ED1 staining were all collected on day 14. We have revised the sentence at Line 400-402.
Ref1: Levkovitch-Verbin, H.; Harris-Cerruti, C.; Groner, Y.; Wheeler, L. A.; Schwartz, M.; Yoles, E., RGC death in mice after optic nerve crush injury: oxidative stress and neuroprotection. Invest Ophthalmol Vis Sci 2000, 41, (13), 4169-74.
Ref2: Ohlsson, M.; Mattsson, P.; Svensson, M., A temporal study of axonal degeneration and glial scar formation following a standardized crush injury of the optic nerve in the adult rat. Restor Neurol Neurosci 2004, 22, (1), 1-10.
Ref3: Tsai, R. K.; Chang, C. H.; Wang, H. Z., Neuroprotective effects of recombinant human granulocyte colony-stimulating factor (G-CSF) in neurodegeneration after optic nerve crush in rats. Exp Eye Res 2008, 87, (3), 242-50.
2. In Figure 4, the staining of ED1 appeared to be nonspecific in tissue sections. Please provide the picture of negative control.
Response2: Thanks for your suggestions. We have redone the ED1 staining and corrected it in the revised manuscript.
3. In Figure 7, the band of Nrf2 showed more protein levels than the loading control of housekeeping gene. Was there the same quantity of proteins loaded in western blotting? The author should check the data.
Response3: Thanks for your detailed inspection. Nrf2 and GAPDH are the same quantity of proteins loading is 30ug, the replaced Nrf2 and GAPDH images are the same as the original one. We already corrected the picture in the revised manuscript.
4. Authors have provided very fewer description in the conclusion. It should be elaborated based on outcomes.
Response4: Thanks for your suggestion, we have revised the conclusions, Line525-530.

Reviewer 2 Report
The manuscript submitted by Chiao-Ling Hsu and coauthors demonstrate the neuroprotective effect of a cyathane diterpenoid Erinacine A in a model of traumatic optic neuropathy. All experimental data are clearly presented and discussed in details. The manuscript can be recommended for publication after minor revision.
Minor comments:
- Please, provide the source of Erinacine A (scheme of the synthesis/extraction or supplier).
- The authors indicate in all figure legends "Data are expressed as mean ± SD" and use box plots. Please, describe what boxes and whiskers mean in this case.
- The typographical errors have to be corrected (diterpe. nes).
Author Response
Point1: Please, provide the source of Erinacine A (scheme of the synthesis/extraction or supplier).
Response1: We have added the information in Line 437-438.Biotech Research Institute, Grap King Bio Ltd., Taoyuan 325002, Taiwan
Point2: The authors indicate in all figure legends "Data are expressed as mean ± SD" and use box plots. Please, describe what boxes and whiskers mean in this case.
Response2: A boxplot is a standardized way of displaying the data set based on the minimum, the maximum, the sample median, and the first and third quartiles. The boxplot mainly describes the interquartile range (IQR) and median. Thank you for your valuable suggestion, we have replaced the boxplot with the bar graph.
Point3: The typographical errors have to be corrected (diterpe. nes).
Response3: Dear reviewer, we have corrected it in my manuscript, thank you for your valuable comments.

Reviewer 3 Report
Hsu et al. reported neuroprotective effects of Erinacine A on a rat model of TON. The study was well designed and conducted. I only have one question. Did authors observe any differences in the latencies in fVEP ?
Author Response
Point1: I only have one question. Did authors observe any differences in the latencies in fVEP ?
Response1: In this study, the latencies of the P1 lantency in the sham, C+PBS-treated, C+standard EA dose-treated, and C+double EA dose-treated groups were 36.1 ± 3.6, 56.4 ± 14.0, 36.6 ± 5.7, and 40.4 ± 8.3 ms, respectively. In comparison with the sham group, the P1 latency did not show significant difference in the C+standard EA dose-treated group and the C+double EA dose-treated group. For the partial injury of optic nerve, the amplitude measured by fVEP is a reflection of axon number excited, therefore, both evaluations either by P1 latency or amplitude of VEP for ON function are used in the literature.
